# Anticancer Activity of Plant Tocotrienols, Fucoxanthin, Fucoidan, and Polyphenols in Dietary Supplements

**DOI:** 10.3390/nu16244274

**Published:** 2024-12-11

**Authors:** Gabriel Lara-Hernández, José Alberto Ramos-Silva, Elvia Pérez-Soto, Mario Figueroa, Ericka Patricia Flores-Berrios, Laura Sánchez-Chapul, José Luis Andrade-Cabrera, Alexandra Luna-Angulo, Carlos Landa-Solís, Hamlet Avilés-Arnaut

**Affiliations:** 1Laboratorio de Biomedicina y Salud Ocupacional, Escuela Nacional de Medicina y Homeopatía, Instituto Politécnico Nacional, Ciudad de México 07320, Mexico; gabobmol@gmail.com (G.L.-H.); elvperezs@ipn.mx (E.P.-S.); 2Facultad de Ciencias Biológicas, Instituto de Biotecnología, Universidad Autónoma de Nuevo León, Campus Ciudad Universitaria, Av. Universidad S/N, San Nicolás de los Garza 66455, Mexico; qfbalbertors@gmail.com; 3Facultad de Química, Universidad Nacional Autónoma de México, Ciudad de México 04510, Mexico; mafiguer@unam.mx; 4Investigación y Desarrollo, Biodesarrollos Valmex, Circuito Crisantemos 10, Tlajomulco de Zúñiga 45640, Mexico; eflores@biovalmex.com; 5Laboratorio de Enfermedades Neuromusculares, División de Neurociencias Clínicas, Instituto Nacional de Rehabilitación “Luis Guillermo Ibarra Ibarra”, Ciudad de México 14389, Mexico; lchapul@yahoo.com (L.S.-C.); lunangulo@gmail.com (A.L.-A.); 6Laboratorio de Patología Clínica, Instituto Nacional de Rehabilitación “Luis Guillermo Ibarra Ibarra”, Ciudad de México 14389, Mexico; jandrade71@yahoo.com; 7Unidad de Ingeniería de Tejidos, Terapia Celular y Medicina Regenerativa, Instituto Nacional de Rehabilitación “Luis Guillermo Ibarra Ibarra”, Ciudad de México 14389, Mexico; cls_73@hotmail.com

**Keywords:** cytotoxicity, apoptosis induction, cancer cell lines, natural products, antioxidants

## Abstract

**Background:** Plants and algae harbor diverse molecules with antioxidant activity and have been demonstrated to directly inhibit cancer cell growth and mitigate the oxidative damage associated with certain antitumor therapies. While antioxidant supplementation, either alone or in combination with chemotherapy, has shown promise in improving quality of life, further research is needed to explore the effects of antioxidant combinations on specific cancer cell lines. **Methods:** In this study, the in vitro cytotoxic and apoptotic properties of natural compounds derived from plants and algae, as well as certain dietary supplements, were investigated against various human cancer cell lines, including bone, leukemia, colorectal, breast, and prostate cancers. **Results:** Apple polyphenols, fucoxanthin, and plant-derived tocotrienols exhibited cytotoxic effects across all lines; however, tocotrienols demonstrated the most potent, time-dependent cytotoxic activity, with a half-inhibitory concentration (IC_50_) of 4.3 μg/mL in bone cancer cells. Analysis of dietary supplements 2.1, 4.0, and 10.0 revealed that supplement 10.0 exhibited specific cytotoxic activity against bone cancer line TIB-223 and colorectal cancer cell line Caco2, with IC_50_ values of 126 μg/mL and 158 μg/mL, respectively. Both tocotrienols and supplement 10.0 induced morphological changes in TIB-223 cells, inhibited cell migration (anti-metastatic activity), and promoted apoptosis, as evidenced by caspase 3/7 activation in both bone and colorectal cancer cells. **Conclusions:** These findings provide valuable insights for the development of targeted dietary supplements to enhance the anticancer effect of conventional chemotherapy in specific cancer types.

## 1. Introduction

Cancer remains one of the leading causes of mortality worldwide. Colorectal cancer ranks as the third most prevalent malignancy and is the second leading cause of cancer-related mortality globally [1]. At the time of diagnosis, over 20% of patients present with distant metastases, and their five-year survival rate is alarmingly low, falling below 15%. Even in cases where metastases are absent, approximately 40% of patients experience disease recurrence after the initial treatment, and chemotherapy remains the cornerstone of therapy for both metastatic and recurrent colorectal cancer [2].

Similarly, malignant fibrous histiocytoma of the bone (MFH) accounts for 2% to 5% of all primary malignant bone tumors, with the femur being the most frequently affected site. Most MFH cases are classified as high-grade tumors, with five-year survival rates ranging between 34% and 50%. Treatment typically involves aggressive surgical resection, combined with chemotherapy [3]. Despite significant advances in antitumor therapies over recent decades, the side effects associated with these treatments continue to pose a major challenge for patients. One of the most prevalent adverse effects is chemotherapy-induced oxidative damage, which can compromise the patient’s quality of life and limit the efficacy of treatment. In this context, antioxidants have emerged as potential adjunctive agents that not only mitigate oxidative damage but may also exert direct antitumor effects [4,5].

Plants and algae are rich sources of bioactive compounds with antioxidant properties, many of which have demonstrated specific cytotoxic effects against tumor cells in preclinical studies [6,7,8,9]. Among these compounds, apple polyphenols, fucoxanthin, and plant-derived tocotrienols have garnered particular attention due to their ability to inhibit cancer cell proliferation and promote apoptosis, positioning them as promising candidates for the development of complementary oncology therapies [10,11,12]. However, despite growing interest in the use of antioxidants in cancer treatment, there is considerable debate regarding their efficacy and safety, especially when used in combination with chemotherapy [13,14]. Some studies suggest that antioxidants may interfere with the effects of chemotherapy, while others advocate for their use, citing benefits in reducing side effects and enhancing antitumor activity [15,16].

In this context, the present study aims to evaluate the cytotoxic and pro-apoptotic properties of natural compounds derived from plants and algae, as well as dietary supplements formulated with these compounds, regarding various human cancer cell lines, including bone cancer, leukemia, colorectal cancer, breast cancer, and prostate cancer. Therefore, we hypothesized that formulations with a combination of natural compounds could synergistically enhance their antitumor potential. Specifically, the study examines the activity of plant-derived tocotrienols, fucoxanthin, fucoidan, astaxanthin, and apple polyphenols, as well as the efficacy of three specific dietary supplements. These compounds and supplements were evaluated for their ability to induce apoptosis, inhibit cell migration, and reduce cell viability in selected cancer cell lines, with the goal of identifying formulations that could enhance the effects of conventional antitumor therapies.

The primary objective of this work is to provide a deeper understanding of how these antioxidants and dietary supplements might be effectively integrated into therapeutic regimens against cancer. The findings not only offer new insights into the potential efficacy of these compounds but also underscore the need for further studies to elucidate their mechanisms of action and their interactions with conventional treatments. Ultimately, this study aims to contribute to the development of more effective and less toxic therapeutic strategies for the treatment of bone and colorectal cancer.

## 2. Materials and Methods

### 2.1. Compounds and Dietary Supplements

All compounds used in this study were purchased from BGG (Beijing, China) https://bggworld.com (accessed on 30 September 2024): AstaZine^®^ (astaxanthin from *Haematococcus pluvialis*), TheraPrime^®^ (palm-rice-annatto tocotrienols), ApplePhenon^®^ (apple polyphenols), ThinOgen^®^ (fucoxanthin from *Laminaria japonica*) and FucoMax^®^ (fucoidan from *Laminaria japonica*, *Undaria pinnatifida*, and *Cladosiphon okamuranus*). The following nomenclatures will be applied in subsequent sections of this manuscript: astaxanthin (AXT), plant-derived tocotrienols (PT3), apple polyphenols (APP), fucoxanthin (FXT), and fucoidan (FUC). These abbreviations will be consistently employed to refer to the corresponding natural compounds in all relevant tables, figures, and in the results and discussion sections to ensure clarity and coherence in presenting the data. The cytotoxic activity on different cell lines was determined for each of the compounds, as well as for the different dietary supplements formulated with the compounds. The dietary supplements were purchased from BioMaussan^®^ (Ciudad de Mexico, Mexico) and contain the following formulation per 100 mL, according to their declared labeling: 2.1 algas marinas premium (astaxanthin and fucoxanthin 1950 mg and apple polyphenols 1250 mg), 4.0 fucoxanthin special formula (fucoxanthin 3330 mg), and 10.0 ultra (fucoidan 7142 mg and tocotrienols 6550 mg). Doxorubicin (DOX; Zytokil, laboratorios PISA, Ciudad de Mexico, Mexico) and paclitaxel (PTX; Ofoxel, laboratorios PISA, Ciudad de Mexico, Mexico) were used as positive controls.

### 2.2. Liquid Chromatography—Mass Spectrometry Analysis

The presence of the ingredients listed on the label of each dietary supplement was analyzed using HPLC-MS (Appendix A). Dietary supplements (3 mg/mL) and compounds (3 mg/mL) were dissolved in dioxane-MeOH-H2O (1:1:1) (Merck, Darmstadt, Germany, Cat. 103132) and analyzed on an Acquity UPLC (Waters Inc., Milford, MA, USA) coupled to a SQD2 (Waters Inc., Milford, MA, USA) mass spectrometer. LC analysis was performed on a Kinetex XB-C18 column (100 mm × 2.1 mm, I.D., 1.7 μm, 100 Å; Phenomenex, Torrance, CA, USA) at 40 °C, with a solvent system of 85:15 H_2_O-CH_3_CN (both phases with 0.1% formic acid) for 0.5 min (isocratic), then to 100% CH_3_CN for 7.5 min, then held for 1.5 min with CH_3_CN, and returned to the starting conditions, with a flow rate of 0.3 mL/min and an injection volume of 7 μL. MS data were obtained using an ESI source (positive and negative modes) at a full scan range (*m*/*z* 100–2000), with the following settings: capillary voltage, 3.5 kV; cone voltage, 35 V; desolvation temperature, 500 °C; desolvation gas flow, 400 L/h; and cone gas flow, 10 L/h. Data were analyzed using the MassLynx v4.1 software (Waters, Inc., Milford, MA, USA).

### 2.3. Cell Lines and Cell Culture

Tumor cell lines, including fibrous histiocytoma (GCT/TIB-223), colorectal cancer (Caco2/HTB-37), breast cancer (MCF-7/HTB-22), bone marrow (K-562/CCL-243), and prostate cancer (DU 145/HTB-81), as well as normal cell lines (Vero (CCL-81) and fibroblasts (Detroit 548/CCL-116)), were cultured in McCoy’s 5A (Gibco, Grand Island, NY, USA, Cat. 16600-082), DMEM (Gibco, Grand Island, NY, USA, Cat. 11995-065)) and RPMI 1640 media (Gibco, Grand Island, NY, USA, Cat. 22400-089) supplemented with 10% (*v*/*v*) FBS (Gibco, Grand Island, NY, USA, Cat. 26140-079). The cells were maintained at 37 °C in a humidified atmosphere with 5% CO_2_. All cell lines were obtained from the American Type Culture Collection (ATCC, Manassas, VA, USA).

### 2.4. Cytotoxicity Assay

The cytotoxic effects were assessed using MTT (3-(4,5-dimethylthiazol-2-yl)-2,5-diphenyltetrazolium bromide) assays, following a method that was previously reported [17]. After an initial 24 h of incubation, the cells were exposed to different concentrations of plant and algae extracts or dietary supplements and incubated for another 24 h. Subsequently, each well received a solution of MTT (Bio Basic Canada Inc., Markham, ON, Canada, Cat. T0793) and the cells were incubated at 37 °C for an additional 4 h. As a control, a medium containing DMSO (Fisher Chemical, Fair Lawn, NJ, USA, Cat. D128-1) was used. Doxorubicin- or paclitaxel-treated cells served as positive controls, while untreated cells functioned as the negative control. The IC_50_ values for each cancer cell line were determined by plotting a non-linear four-parameter regression curve [18]. Each compound concentration was tested independently in triplicate, with three technical replicates per assay. To evaluate the time-dependent response of PT3 and supplement 10.0, TIB-223 cells were exposed to each compound at their respective IC_50_ concentrations. Cell viability was assessed using the MTT assay at 12, 24, 36, and 48 h post-treatment. Doxorubicin and paclitaxel were included as positive controls to validate the experimental conditions. For each experimental condition, three replicates were conducted, and the entire assay was repeated across three independent experiments to ensure reproducibility and statistical reliability.

### 2.5. Cell Morphology

Cells were plated at a density of 5 × 10^5^ in a 12-well plate and allowed to incubate for 24 h. After attachment, the cells were treated with each compound (PT3, FXT, APP, FUC, and AXT) using the IC_50_ concentration at 12 and 24 h. After each incubation period, an inverted microscope (Olympus IX71, Tokyo, Japan), coupled with an Infinity 1–2 camera from Lumenera (Ottawa, ON, Canada), was used to capture photographs and observe changes in cell morphology. PBS-treated cells were used as a control. Three technical replicates were performed for each of the three independent experiments.

### 2.6. Wound and Healing Assay

The wound-healing assay was performed, following an established protocol [19,20]. Cells were plated in 12-well plates at a density of 5 × 10^4^ to 6 × 10^5^ cells per well and allowed to grow overnight until they reached full confluence. A scratch was created in the cell monolayer using a pipette tip, after which the wells were rinsed with PBS (Cellgro, Manassas, VA, USA, Cat. 21-040-CV) to eliminate detached cells. The remaining cells were then exposed to their respective IC_50_ concentrations for 24 and 48 h. At the end of the incubation period, the cells that migrated into the scratched area were photographed every 24 h until the wound was completely closed in the control cells. After that, every image was analyzed with TScratch software version 1.0 [21]. The mean percentage of injured areas was calculated for each field. A total of three independent experiments were conducted, each including three technical replicates.

### 2.7. Dual Acridine Orange/Ethidium Bromide (AO/EB) Fluorescent Staining

The AO/EB double staining assay was performed, as previously described [22]. For the AO/EB staining, 4 × 10^4^ cells were seeded in a 96-well plate (Thermo Fisher Scientific, Waltham, MA, USA, Cat. 167008). Cells were treated with PT3, the dietary supplement 10.0, and DOX at their corresponding IC_50_ concentrations for 4 h. Then, a dual fluorescent staining solution containing 100 µg/mL AO and 100 µg/mL EB (AO/EB, Sigma-Aldrich, St. Louis, MO, USA, Cat. 318337 and E7637) was added to each well, as described previously [23]. The cells were viewed and counted using the EVOS M5000 fluorescence microscope (Thermo Fisher Scientific, Waltham, MA, USA) at 20× magnification, with an excitation filter at 375 nm (green filter) and red filter at 480/560 nm, respectively. Three fields per sample were examined according to the following criteria: live cells (with a green and circular nucleus, uniformly distributed in the center of the cell), early apoptotic cells (nucleus with chromatin condensation showed yellow-green fluorescent by AO staining), late apoptotic cells (the nucleus with chromatin condensation showed orange fluorescence with EB staining) and necrotic cells (cell volume had increased, showing red fluorescence). The AO/EB staining method was repeated three times.

### 2.8. Caspase Assay

Cells were seeded, treated with PT3, DOX, PTX, and the dietary supplement 10.0 at their respective IC_50_ values, and incubated for 4 h. Caspase activity was determined using CellEvent™ Caspase-3/7 Green ReadyProbes™ Reagent (Invitrogen, Carlsbad, CA, USA Cat. R37111) according to the manufacturer’s instructions, with an inverted microscope with fluorescence (EVOS™ M5000 Imaging System, Thermo Fisher Scientific, Waltham, MA, USA). Cells were viewed and counted at a 20× magnification with an excitation filter at 375 nm (green filter). Three separate experiments were conducted, with each experiment comprising three technical replicates.

### 2.9. Determination of Total Antioxidant Capacity (TAC) of Natural Compounds and Dietary Supplements

A FRAP assay was performed to determine the reducing capacity of natural compounds (BGG) and dietary supplements (BioMaussan^®^, Mexico city, Mexico) in a redox reaction, as previously reported [24]. The D-α-tocopherol succinate (positive control) was purchased from Sigma-Aldrich, St. Louis, MO, USA, Cat. 47782. For the determination of FRAP, samples of the natural compounds were diluted in deionized water at different concentrations: AXT, 50 mg/mL (1:100); FUC, 33.2 mg/mL (1:200); PT3, 5 mg/mL; APP, 5 mg/mL (1:100), and FUC, 50 mg/mL (1:10), depending on the content relative to dietary supplements. The supplements were diluted in deionized water at different ratios and annotated as follows: 2.1 (1:250); 4.0 (1:50), and 10.0 (1:100). Then, the absorbance was read at 595 nm. The standard curve was constructed using a Trolox solution (range 4.68–300 µM), and the results were corrected for dilution and are expressed as mM Trolox equivalent antioxidant capacity per milliliter (TEAC/mL). All solutions were freshly prepared and used immediately. Measurements were performed in triplicate.

### 2.10. Statistical Data Analysis

The data obtained from the MTT assay (IC_50_), AO/EB staining, wound-healing assays, caspase activity measurements, and antioxidant capacity were reported as the mean ± SEM from three independent experiments. Statistical analysis was conducted using GraphPad Prism software (version 8.0.1) [25], employing a one-way analysis of variance (ANOVA) with Tukey’s test for subsequent multiple comparisons. A significance threshold of *p* < 0.05 was established.

## 3. Results

### 3.1. Cytotoxic Activity of Natural Compounds and Dietary Supplements on Different Human Cancer Cell Lines

The cytotoxicity assay (MTT) was performed to assess the effect of natural plant and algal compounds and dietary supplements on cancer cell lines. This experiment aimed to evaluate the efficacy of plant tocotrienols (PT3), fucoxanthin (FXT), apple polyphenols (APP), fucoidan (FUC), and astaxanthin (AXT), as well as two commercially available antitumor agents, doxorubicin (DOX) and paclitaxel (PTX), in reducing cell viability, which could provide insights into their potential as anticancer agents. IC_50_ values represent the concentration of each compound or dietary supplement required to reduce the growth of 50% of the cell population (see Table 1 and Table 2). PT3 and FXT exhibited the most potent cytotoxic effects in all tested cancer cell lines, but PT3 have relative selectivity for cancer cells, along with reduced toxicity toward normal cells (Vero and Detroit 548 cells).

Based on the results summarized in Table 1, PT3 demonstrated the strongest cytotoxic activity on TIB-223 cells with an IC_50_ value of 4.3 μg/mL, which was more potent than FXT (14 μg/mL) and APP (43 μg/mL). In contrast, FUC and AXT exhibited minimal cytotoxicity as their IC_50_ values exceeded 1000 μg/mL. DOX and PTX, used as positive controls, showed remarkable potency, with IC_50_ values of 1.16 μg/mL and 1.31 μg/mL, respectively, further highlighting their established effectiveness as anticancer agents.

These findings indicate that PT3 exhibit substantial cytotoxicity that is comparable to standard chemotherapeutic agents, while FXT and APP show moderate activity. FUC and AXT, on the other hand, appear to be much less effective under the above test conditions.

According to the results in Table 2, supplement 10.0 demonstrated the highest cytotoxic activity on TIB-223 cells, with an IC_50_ of 126 μg/mL, which was significantly lower than supplements 2.1 (1160 μg/mL) and 4.0 (2302 μg/mL), indicating that 10.0 was the most potent among the tested supplements in the bone cancer cell line. Supplement 10.0 also demonstrated significant cytotoxic activity in colorectal cancer cells, with an IC_50_ value of 158 μg/mL. These findings contrast with the higher IC_50_ values of supplement 10.0 observed in non-tumor cell lines. This suggests that the supplement exhibits a reduced impact on normal cells, highlighting its potential for targeted therapy. Conversely, supplements 2.1 and 4.0 exhibited limited cytotoxicity under the conditions tested. Additionally, an MTT assay was performed to evaluate the time-dependent response of TIB-223 cells to treatment with PT3 and supplement 10.0 from 0 to 48 h, in which it was observed that cytotoxic activity is time-dependent for PT3, DOX, and PTX, while dietary supplement 10.0 appears to have a less pronounced cytotoxic effect over time (Appendix A).

### 3.2. Morphological Changes in a Human Cancer Cell Line Exposed to Natural Compounds from Plants and Algae

Given that the results from Table 1 indicated that the TIB-223 bone cancer cell line exhibited the highest sensitivity to the cytotoxic effects of the natural compounds, we decided to assess the morphological alterations of these cells in response to the five ingredients utilized in the formulation of various dietary supplements. Morphological changes in human cancer cells were monitored for up to 24 h. Treatment of TIB-223 cells with the tested ingredients revealed significant morphological changes from a fibroblastic to a spherical shape, which is indicative of potential cytotoxic effects and reduced adhesion capacity. At 12 h of incubation, cells exposed to PT3 and FXT exhibited thinner plasma membranes with poorly defined edges (Figure 1). The percentage of spherical cells increased notably after 24 h of treatment with PT3 and FUC, reaching almost 90% and 70%, respectively, compared to the control, where less than 10% of cells exhibited this change (Figure 2). FXT also demonstrated a significant effect, with approximately 50% of the cells becoming spherical after 12 and 24 h of treatment. The response to APP and AXT was less pronounced, with around 20–30% of cells showing a change in morphology at the same time points.

These findings suggest that PT3, FXT, and FUC exhibit considerable cytotoxic effects, leading to cell rounding and possible detachment from the culture substrate, a hallmark of decreased cell adhesion and viability. The induction of a spherical shape could be linked to the disruption of the cytoskeleton or apoptosis, processes that are commonly associated with reduced adhesion and cell death in cancer cells. The strong morphological response observed with PT3 and FUC highlights their potential as therapeutic agents, targeting cellular integrity and survival in bone cancer cells. Given that there was no vacuolation observed in the cytoplasm of the cells, we determined that autophagy is not a factor in the mechanism of cell death. Notably, the control cells (untreated) showed no significant morphological changes during the 24-hour period and exhibited a substantial increase in number.

### 3.3. Migration of Bone Cancer Cells In Vitro

To assess the impact of PT3 and supplement 10.0 on the migration of TIB-223 bone cancer cells, a wound-healing assay was conducted (Figure 3). The progress of the cancer cells was observed until the artificially created wound had healed completely. The untreated cells showed full closure of the wound after 48 h and achieved over 50% closure by 24 h (Figure 4). In contrast, TIB-223 cells displayed a notable reduction in wound healing when exposed to PT3 or supplement 10.0, with the wound remaining open even after 48 h. Doxorubicin was used as a positive control in the wound-healing assay, leading to an enlargement of the wound area as a result of considerable detachment of the bone cancer cells (Figure 3 and Figure 4). These findings indicate that PT3 and supplement 10.0 inhibit and delay the migration of TIB-223 cells in vitro.

Cell migration assays were also performed on the Caco2 colorectal cancer cell line, given the epidemiological significance of this disease and the line’s sensitivity to dietary supplement 10.0 (which exhibited the second-lowest IC_50_ after TIB-223 cells, as shown in Table 2). In addition to the two cancer cell lines, a normal cell line (Detroit 548) was included in the assay as a reference. By the end of the experiment, PT3 and supplement 10.0 maintained a wound closure performance of over 20% in Caco2 cells, whereas in fibroblasts, wound closure remained closer to 10% (Appendix A). These results suggest that PT3 and supplement 10.0 delay the migration of cancer cells more effectively than normal cells.

From this point onward, all cell experiments were conducted on both cancer cell lines (TIB-223 and Caco2) and the normal cell line (Detroit 548).

### 3.4. Dual Acridine Orange/Ethidium Bromide (AO/EB) Fluorescent Staining

To evaluate the type of cell death, we employed acridine orange/ethidium bromide (AO/EB) fluorescent staining to observe changes indicative of apoptosis in TIB-223 cells. AO, a dye that easily penetrates cell membranes, binds to the nucleic acids of living cells. In contrast, EB does not cross membranes but can easily enter non-viable cells, attaching to their DNA. When both AO and EB are applied together, viable cells exhibit green fluorescence, while non-viable cells show red fluorescence under a fluorescence microscope. After a 4-hour treatment with PT3 or DOX as a positive control, bone cells were labeled using the AO/EB method. The dual staining was examined under a fluorescent microscope (Figure 5) and demonstrated that treatment with PT3 induced a greater number of dead cells (red color) compared to control cells; however, DOX treatment was much more potent, causing death in the majority of cancer cells. These results suggest that PT3 may induce the death of bone cancer cells through the mechanism of apoptosis.

The results shown in Figure 6 reveal significant differences in cell viability and apoptosis across the treatments. The control group displayed the highest percentage of live cells (~90%) with the minimal induction of apoptosis or cell death. Treatment with PT3 led to a marked reduction in live cells (30%) and a substantial increase in late apoptotic cells (~30%) and dead cells (20%), suggesting a potent pro-apoptotic effect. The dietary supplement 10.0 maintained a higher proportion of live cells (~60%) compared to PT3, though it still induced early and late apoptosis (10%), reflecting moderate cytotoxicity. Doxorubicin, as expected, exhibited a strong cytotoxic profile, with a significant increase in dead cells (~40%) and a reduction in live cells.

In summary, the AO/EB staining assay demonstrated that both PT3 and DOX are potent inducers of apoptosis in TIB-223 cells, with DOX showing the strongest cytotoxic effect. The 10.0 supplement displayed milder apoptosis induction, suggesting lower cytotoxicity compared to the other treatments.

A comparison of the dual fluorescent staining results across the three cell lines indicates that both PT3 and supplement 10.0 selectively induce apoptosis in cancer cell lines (Caco2 and TIB-223) while exhibiting greater preservation of the normal fibroblasts compared to doxorubicin (Appendix A). These findings underscore the potential of PT3 and supplement 10.0 as safer alternatives for targeting cancer cells since they demonstrate lower cytotoxicity toward normal cells relative to DOX.

### 3.5. Caspase Activity in Cancer Cells Treated with Plant Tocotrienols and Food Supplement 10.0

Caspases 3 and 7 play a crucial role as executioner enzymes in the process of apoptosis. To validate the induction of apoptosis in cancer cells by PT3 and the dietary supplement 10.0, we measured the activity of these caspases in bone and colon cancer cells, as well as in the normal fibroblast. Treatment of TIB-223 (Figure 7), Caco2 (Appendix A), and Detroit 548 cells (Appendix A) with PT3 and supplement 10.0 resulted in the detectable activation of caspases 3 and 7 (indicated by green fluorescence signals) compared to the untreated control cells. Collectively, these findings provide strong evidence that PT3 and supplement 10.0 exert cytotoxic effects through the activation of apoptosis, which is mediated by caspases 3 and/or 7.

The results from the caspase 3/7 activity assay, shown in Figure 8, indicate the significant induction of caspase activity following PT3 treatment, with the highest percentage of caspase 3/7-positive cells (22%). This suggests that PT3 treatment strongly induces apoptosis via the caspase pathway in bone cancer cells. PT3 also induced caspase 3/7 activity in Caco2 and Detroit 548 cells (Appendix A). In contrast, treatment with 10.0, DOX, and PTX resulted in a moderate increase in caspase activity, with similar percentages of caspase 3/7-positive cells (6–8% in Figure 8). These results indicate that while these treatments also activate apoptosis, they do so at a lower level compared to PT3. The lower number of cells with activated caspases in treatments with DOX and PTX could be associated with the fact that caspase activation times for commercial antitumor drugs may be less than 4 h. The control group showed minimal caspase 3/7 activation (2%), confirming the low baseline of apoptotic activity in untreated cells. In Caco2 cells (Appendix A), PT3 and 10.0 both significantly increased caspase 3/7 activity compared to the control, and DOX and PTX exhibited a lower level of induction than PT3 and 10.0. The caspase activity assay highlights the potent apoptotic effect of PT3 and supplement 10.0 on TIB-223 and Caco2 cells, driven by the significant activation of caspase 3/7.

### 3.6. Total Antioxidant Capacity (TAC) of Natural Compounds and Dietary Supplements According to the FRAP Assay

A ferric reducing antioxidant power (FRAP) assay was conducted to evaluate the total antioxidant activity of various natural compounds and food supplements. The FRAP assay measures the reducing potential of the antioxidants present in the samples by their ability to convert ferric (Fe^3+^) to ferrous (Fe^2+^) ions. The tested compounds included AXT, FXT, APP, PT3, and FUC, as well as the different formulations (2.1, 4.0, 10.0) of a dietary supplement.

The results shown in Figure 9 indicate a significant variation in antioxidant activity across the compounds. Formulation 2.1 exhibited the highest antioxidant activity (~150 mM TEAC/mL), significantly outperforming all other compounds. PT3 demonstrated moderate antioxidant activity (66 mM TEAC/mL), which was comparable to that of FUC (43 mM TEAC/mL) and the 10.0 supplement (87 mM TEAC/mL) but was significantly lower than formulation 2.1. AXT, FXT, and 4.0 displayed relatively low antioxidant activity, similar to that in the control group (α-tocopherol at 1 mg/mL).

In summary, the FRAP assay revealed that formulation 2.1 possesses the strongest antioxidant capacity among the tested compounds, while PT3 and the 10.0 supplement also demonstrated significant antioxidant potential. These findings suggest that these natural compounds can play a role in reducing oxidative stress through their antioxidant properties.

## 4. Discussion

The results of this in vitro study provide significant evidence of the cytotoxic and apoptotic potential of natural compounds derived from plants and algae, as well as of dietary supplements formulated with these compounds, against various cancer cell lines. MTT assays revealed that PT3 and FXT exhibited the most potent cytotoxic effects across all tested cancer cell lines, including bone, leukemia, colon, breast, and prostate cancers. Notably, the strongest effect was observed in the TIB-223 bone cancer cell line, with an IC_50_ of 4.3 μg/mL.

Tocotrienols, a less common form of vitamin E, have been reported to inhibit cell proliferation, induce apoptosis, suppress angiogenesis, and reduce cancer cell migration [26,27]. The particularly high susceptibility of the TIB-223 cell line to PT3 suggests that these compounds could be especially effective in treating bone cancers, which is consistent with other studies that have found similar effects in different cancer models. Although the precise mechanism by which tocotrienols exert their cytotoxic effects is not directly addressed in this study, the existing evidence suggests that tocotrienols may induce oxidative stress, activate the apoptotic signaling pathways, inhibit the activation of transcription factors such as NF-κB, and suppress growth receptor signaling [27]. Additionally, tocotrienols appear to have relative selectivity for cancer cells, with reduced toxicity toward normal cells [26], and have a synergistic anticancer effect when combined with chemotherapeutic agents such as gefitinib [28].

Fucoxanthin is a carotenoid that is primarily found in brown algae. Previous studies have demonstrated its antitumor properties, including the induction of apoptosis, inhibition of cell proliferation, and suppression of metastasis [29,30,31,32] through mechanisms involving oxidative stress induction and the activation of apoptotic pathways [33]. The findings of this study support these observations, indicating that fucoxanthin is effective against a broad range of cancer cell lines, although with slightly less potency compared to PT3. While in vitro cancer cell line models are valuable for assessing direct cytotoxic effects, they do not fully capture the complexity of the in vivo tumor microenvironment. Therefore, future research should focus on evaluating the effects of PT3 and FXT in animal models of cancer, either alone or in combination with chemotherapy.

When plant- and algae-derived compounds were evaluated in dietary supplement formulations (2.1, 4.0, and 10.0), the 10.0 formulation demonstrated remarkable cytotoxic efficacy against all tested cell lines, particularly against the TIB-223 and Caco2 cancer cell lines, with IC_50_ levels of approximately 126 μg/mL and 158 μg/mL, respectively. In contrast, supplement 10.0 showed higher IC_50_ levels in the normal cell lines Detroit 548 and Vero. These findings suggest that the 10.0 supplement (containing FUC and PT3) could be a promising candidate for the development of adjuvant therapies in bone and colorectal cancer treatment.

The low IC_50_ value observed for the 10.0 supplement indicates that PT3 may be the primary contributors to the observed cytotoxic activity, which is consistent with previous studies that have demonstrated their high efficacy against bone (SW1353) [34], breast, ovary, liver, colon, pancreas, lung (A549), glioblastoma (U87MG) [35] and prostate cancer cells [36]. Although FUC, another key component of the 10.0 supplement, exhibited IC_50_ values exceeding 1000 μg/mL in this study, its presence in the formulation should not be dismissed as irrelevant. There is evidence suggesting that FUC may act synergistically with chemotherapy agents (cytarabine, cisplatin, tamoxifen, paclitaxel, lapatinib, and cetuximab) and other compounds, such as tocotrienols, to enhance their anticancer effects [12,37,38,39,40]. Therefore, it is possible that the combination of FUC and PT3 in the 10.0 supplement contributes to a more potent cytotoxic effect than would be expected from these compounds individually, although this synergistic effect warrants further investigation. Fucoidan has been shown to inhibit the proliferation of Caco2 cells in a dose-dependent manner. Fluorescent staining assays using Hoechst and Annexin V/PI have confirmed that fucoidan induces apoptosis in these cells. Additionally, fucoidan treatment has been associated with increased reactive oxygen species (ROS) production and enhanced mitochondrial membrane permeability in Caco2 cells [41]. Similarly, the fucoidan extracted from *Sargassum cinereum* has demonstrated cytotoxic effects on the HCT-15 colon cancer cell line [42].

In other studies, fucoidan caused cell-cycle arrest at the G1 phase in a chemo-resistant non-small-cell bronchopulmonary carcinoma cell line. This effect was accompanied by the significant downregulation of cyclin D1, cyclin D2, and CDK4, suggesting that the arrest is mediated by the suppression of cyclin-dependent kinase (CDK) activity. This suppression is likely due to the direct binding of fucoidan to CDKs 2 and 4 [37]. The antitumor properties and molecular mechanisms of fucoidan have been further validated in non-small-cell lung cancer models, where it inhibited angiogenesis via the mTOR signaling pathway. Fucoidan also promoted apoptosis in these cells by increasing the Bax/Bcl-2 ratio, highlighting its potential as an effective therapeutic agent [43].

The variable performance of the other evaluated dietary supplements (2.1 and 4.0) also provides interesting insights. The 2.1 supplement (containing AXT, FXT, and APP) exhibited good cytotoxic activity against the Caco2 and K562 cell lines, which may suggest the presence of specific bioactive compounds with a higher affinity for these cell lines. Astaxanthin, a lipid-soluble carotenoid, demonstrates antioxidant activity significantly surpassing that of vitamin E and is approximately 10 times more potent than other carotenoids, including β-carotene. Studies have shown that Caco2 cells exhibit an altered morphology, becoming more rounded, and show a substantial reduction in viability to around 35% when exposed to astaxanthin concentrations of 5–10 mg/mL [44]. Additionally, prior research indicates that fucoxanthin possesses anticancer properties, including the ability to induce DNA fragmentation in Caco2 cells [45,46]. Similarly, it has been reported that apple polyphenols can modulate protein kinase C activity and trigger apoptosis in these cells [47]. Taken together, these findings suggest that the cytotoxic effects observed with Supplement 2.1 are likely attributable to the synergistic action of the diverse natural compounds within its formulation.

Conversely, the lower cytotoxic activity observed in the 4.0 supplement may indicate a less effective formulation or the presence of compounds that are not particularly active against the cell lines tested in this study.

Identifying dietary supplements like the 10.0 formulation, which contains compounds with potent anticancer properties, opens the possibility of using these products as adjuvants in chemotherapy, particularly in patients who poorly tolerate conventional treatments or who require a more personalized approach. Further studies could include transcriptomic and proteomic analyses to understand how these compounds modulate the cellular pathways in different tumor lines. Finally, the variability in the response of different cell lines to the various supplements suggests an opportunity to develop specific formulations for different types of cancer, based on the molecular characteristics of each tumor. This personalized approach could significantly improve clinical outcomes and the quality of life of cancer patients.

The findings of this study highlight the potential of PT3, FXT, and APP, not only as effective cytotoxic agents but also as modulators of cell morphology and adhesion in bone cancer cells [48]. The observed transformation in cell morphology from an elongated, fibroblast-like shape to a rounded, spherical form, accompanied by the loss of cell adhesion, is a clear indication that the cells are undergoing changes associated with apoptosis. These results are consistent with previous studies that have documented the ability of PT3, FXT, and APP to induce apoptosis in various cancer cell lines [49].

To assess whether the cytotoxic activity observed in TIB-223 and Caco2 cells treated with PT3 or supplement 10.0 could affect tumor cell migration in vitro, a wound-healing assay was conducted. Cell migration is a crucial process in both wound healing and metastasis [50]. In the wound-healing assay, a “wound” is created in a cell monolayer, prompting migratory cells to move toward the injured area. This phenomenon is analogous to that occurring during metastasis, where cancer cells migrate from the primary tumor to other tissues. The assay allows for the observation and quantification of cell migration, which can serve as an indicator of the anti-metastatic potential of compounds on cancer cells [50,51]. The fact that untreated TIB-223 and Caco2 cells completely closed the wound within 48 and 72 h, while those treated with PT3 and supplement 10.0 kept a significant portion of the wound open, indicates that both treatments effectively inhibit cell migration. The greater inhibition observed with supplement 10.0 (20% of the wound area remaining open) compared to PT3 alone (10%) is particularly noteworthy, as it supports the hypothesis of a synergistic effect between PT3 and FUC, the main components of supplement 10.0.

This synergistic effect may be explained by the combination of the previously documented anti-metastatic properties of tocotrienols with the modulatory effects of fucoidan on cell migration [52]. Tocotrienols have been reported as potent inhibitors of cell migration and invasion in various cancer types through the modulation of signaling pathways such as PI3K/AKT and inhibition of the expression of matrix metalloproteinases (MMPs), which are essential for extracellular matrix degradation and tumor invasion [53]. In contrast, fucoidan, a sulfated polysaccharide that is present in brown algae, reduced the metastasis of tumor cells to the lungs in animal models that were intravenously injected with rat mammary adenocarcinoma 13762 MAT cells [54]. Initial research demonstrated that fucoidan inhibits cell invasion by competing with tumor cells for binding to laminin, a key component of the basement membrane. Later investigations revealed that fucoidan interacts with fibronectin, thereby preventing tumor cell adhesion and significantly reducing the migration of human breast adenocarcinoma cells [55]. Similarly, low-molecular-weight fucoidan has been found to enhance the suppressive effects of fluoropyrimidine-based chemotherapy on the migration of Caco2 cells, as evidenced by wound and healing assays [56].

PT3 and supplement 10.0 (formulated with PT3 and FUC) induced apoptosis in TIB-223 and Caco2 cancer cells. Staining with acridine orange/ethidium bromide (AO/EB) revealed a significant increase in apoptosis in cells treated with both compounds, which was further confirmed by fluorescence assays demonstrating the activation of caspases 3 and/or 7. These findings highlight the therapeutic potential of PT3 and their combination with FUC in inducing programmed cell death in colorectal and bone cancer cells.

Tocotrienols have been shown to induce apoptosis across various cancer cell lines, including breast and prostate cancer cells, primarily through the activation of caspases 3 and 7 [4,36,57,58,59]. The activation of these caspases is a critical step in the apoptotic signaling pathway, suggesting that the mechanisms of action of tocotrienols may be consistent across different types of tumor cells. Specifically, the activation of caspases 3 and 7 indicates the involvement of the intrinsic apoptotic pathway, which is typically triggered by oxidative stress and DNA damage—mechanisms that have been widely associated with the effects of tocotrienols.

Fucoidan, on the other hand, has been shown to enhance the efficacy of conventional therapies by inducing apoptosis and modulating the immune response [60]. The pro-apoptotic activity of fucoidan, derived from the brown seaweed *Sargassum cinereum,* against Caco2 cells was confirmed through AO/EB staining [41]. Additionally, treatment with fucoidan was shown to elevate the levels of cleaved caspases-8, -9, -7, and -3 in HT-29 colon cancer cells, further supporting its role in inducing apoptosis [61].

The combination of PT3 and FUC in our study suggests a promising approach for cancer treatment, as both compounds appear to act synergistically to increase apoptosis in TIB-223 and Caco2 cells, similar to the findings previously reported for hepatocarcinoma and breast cancer cells [28,62,63,64].

The relationship between antioxidant activity and the antitumor efficacy of natural compounds has been the focus of numerous studies. In this work, we assessed the total antioxidant activity of various formulated supplements, as well as individual compounds derived from plants and algae, using the FRAP assay. The results indicated that dietary supplement 2.1, which contains AXT, FXT, and APP, exhibited the highest total antioxidant activity. However, this same supplement demonstrated lower cytotoxic activity against TIB-223 and Caco2 cells compared to supplement 10.0, which contained PT3 and FUC. This finding is particularly significant as it suggests that high antioxidant capacity does not necessarily correlate with enhanced anticancer efficacy.

This result may be explained by the multifactorial nature of the anticancer mechanisms of natural compounds. While antioxidants can neutralize reactive oxygen species (ROS) and reduce oxidative damage, which may prevent cancer initiation, the induction of apoptosis in cancer cells often involves additional mechanisms, such as the modulation of cellular signaling pathways and the induction of endoplasmic reticulum stress [65]. In this context, PT3, which demonstrated both the highest antioxidant activity and the greatest cytotoxicity among the individual compounds, may be acting through multiple pathways to induce cell death in TIB-223 and Caco2 cells.

While antioxidant supplementation has shown benefits in improving the quality of life for cancer patients, emerging research indicates that its use must be carefully assessed to avoid potential interference with the efficacy of chemotherapy [66]. Future studies should prioritize clinical trials that evaluate the combined use of these dietary supplements with standard treatments and focus on identifying response profiles across various cancer types. Research into the synergistic effects of antioxidant combinations with compounds targeting different cellular pathways could provide valuable insights for the formulation of adjunctive dietary supplements in cancer therapy.

## 5. Conclusions

This study highlights the potential of natural compounds derived from plants and algae, along with dietary supplements, as promising agents for augmenting cancer therapies. Among the tested compounds, tocotrienols exhibited the most potent cytotoxic activity, particularly against bone cancer cells. Furthermore, dietary supplement 10.0 demonstrated selective cytotoxicity against bone and colorectal cancer cell lines, coupled with anti-metastatic properties and apoptosis induction through caspase 3/7 activation. These results underscore the therapeutic potential of tocotrienols and specific supplement formulations as complementary strategies to conventional chemotherapy, particularly in enhancing anticancer treatment efficacy against certain cancer types. Future research should focus on exploring their application in vivo, paving the way for the development of targeted dietary interventions in oncology.

## Figures and Tables

**Figure 1 nutrients-16-04274-f001:**
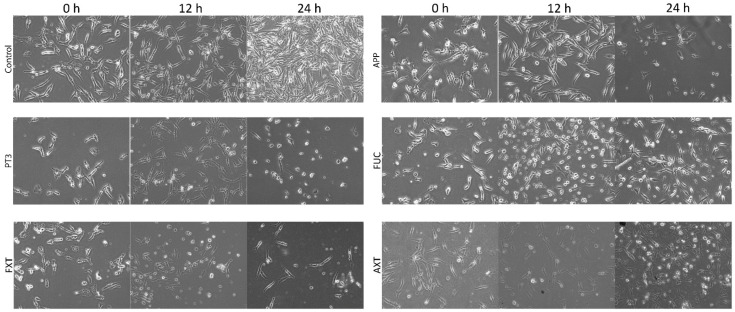
Effects of natural compounds on the morphology of bone cancer cells (TIB-223). Cancer cells were treated with the corresponding IC_50_ value for each compound for 12 or 24 h. The representative results from three independent experiments are shown. Key: PT3 (plant tocotrienols), FXT (fucoxanthin), APP (apple polyphenols), FUC (fucoidan), AXT (astaxanthin), and Control (untreated cells).

**Figure 2 nutrients-16-04274-f002:**
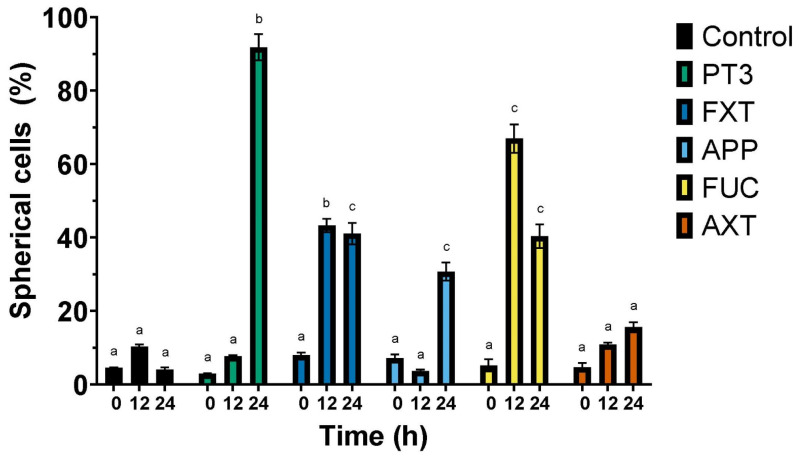
Morphological change from a fibroblastic to spherical shape in bone cancer cells (TIB-223). Cells were treated with the IC_50_ concentration of each compound for 0, 12, or 24 h. The percentage of spherical cells is shown. Key: PT3 (plant tocotrienols), FXT (fucoxanthin), APP (apple polyphenols), FUC (fucoidan), AXT (astaxanthin), and Control (untreated cells). Results are presented as the mean ± standard error of three independent experiments. Bars showing different letters (a–c) at the same time points indicate significant differences, as determined by a one-way ANOVA followed by Tukey’s post hoc test for multiple comparisons. A significance level (*p*-value) of <0.05 was considered statistically significant.

**Figure 3 nutrients-16-04274-f003:**
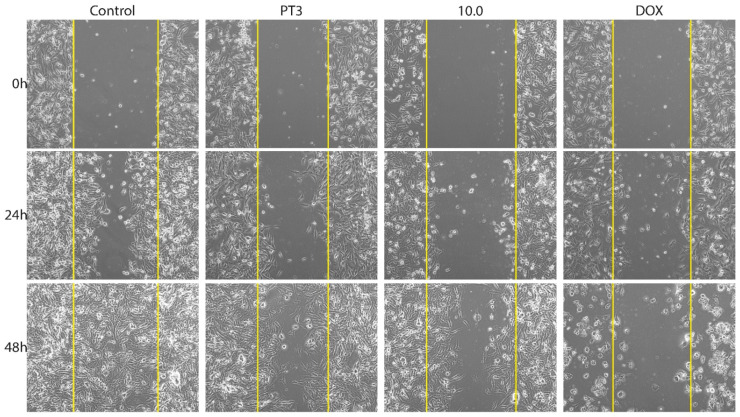
Wound-healing assay using TIB-223 cells. Images depict the monolayer of cancer cells post-wounding, following treatment with plant tocotrienols (PT3), supplement 10.0, and doxorubicin (DOX; positive control) at their respective IC_50_ values, shown after 24 and 48 h. Untreated cells served as a negative control. The yellow lines indicate the initial wound edges at 0 h. The results presented here are representative of three independent experiments.

**Figure 4 nutrients-16-04274-f004:**
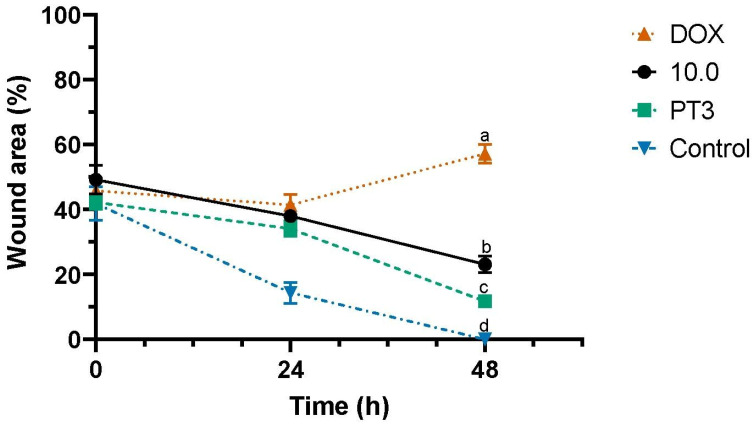
Changes in wound area in the “Wound and Healing” assay of bone cancer cell lines. TIB-223 cells were treated for 24 or 48 h with plant tocotrienols (PT3), supplement 10.0, doxorubicin (DOX), and untreated cells (control). Results are presented as the mean ± standard error of three independent experiments. ^a–d^ Values with different letters at 48 h indicate significant differences, determined by a one-way ANOVA, followed by Tukey’s post hoc test for multiple comparisons. A significance level (*p*-value) < 0.05 was considered statistically significant.

**Figure 5 nutrients-16-04274-f005:**
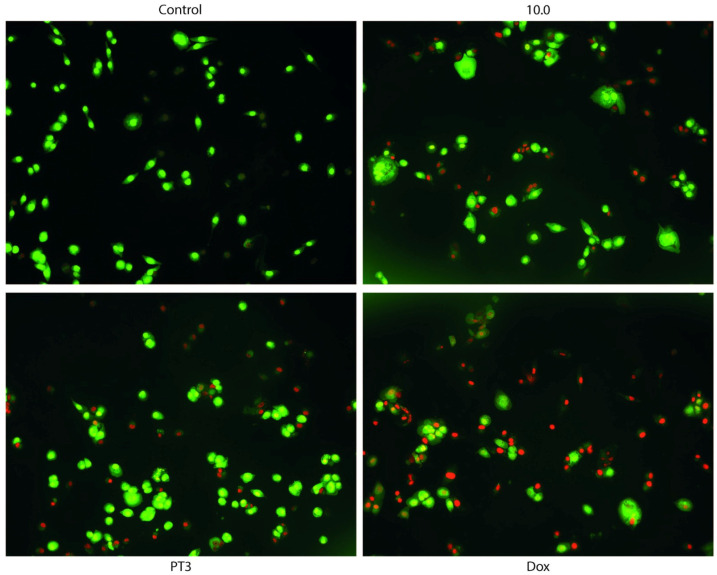
Representative micrographs of AO/EB staining of TIB-223 cells at 10X magnification. Control cells showed a greater number of living cells (green) with the circular nucleus uniformly distributed in the center of the cells. Cells treated with plant tocotrienols (PT3) showed apoptotic cells with the nucleus shown in yellow-green fluorescence by AO staining. The group treated with doxorubicin (DOX) showed a greater number of cells with red nuclei.

**Figure 6 nutrients-16-04274-f006:**
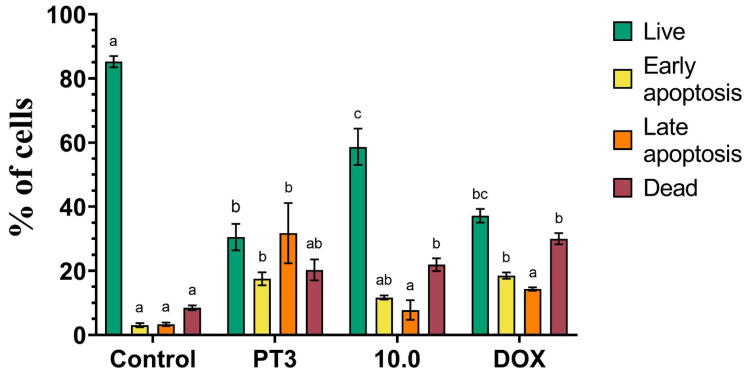
Induction of apoptosis in TIB-223 cells treated with tocotrienols (PT3), 10.0 (dietary supplement), and doxorubicin (DOX). The graph shows the percentage of live cells (green), early apoptotic cells (yellow), late apoptotic cells (orange), and dead cells (red) under different treatments, with their corresponding IC_50_ values. Error bars represent the mean ± SEM from three separate experiments. Bars that have the same color but show different letters (a–c) indicate significant differences, determined by a one-way ANOVA followed by Tukey’s post hoc test. A significance level (*p*-value) of <0.05 was considered statistically significant.

**Figure 7 nutrients-16-04274-f007:**
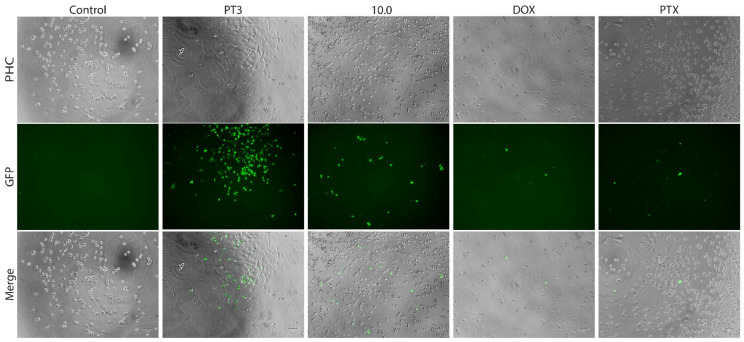
Caspase activity, induced by plant tocotrienols (PT3) and supplement 10.0 in bone cancer cells (TIB-223). The activity of caspases 3 and 7 was determined by fluorescence microscopy. Phase contrast (PHC), GFP filter (GFP), and overlapping images (Merge) are shown. The green fluorescent dots correspond to cells with active caspase-3/7, indicating apoptosis. Doxorubicin (DOX) and paclitaxel (PTX) were evaluated as positive controls. Untreated cells served as the negative control. Each image represents a typical outcome from three separate experiments.

**Figure 8 nutrients-16-04274-f008:**
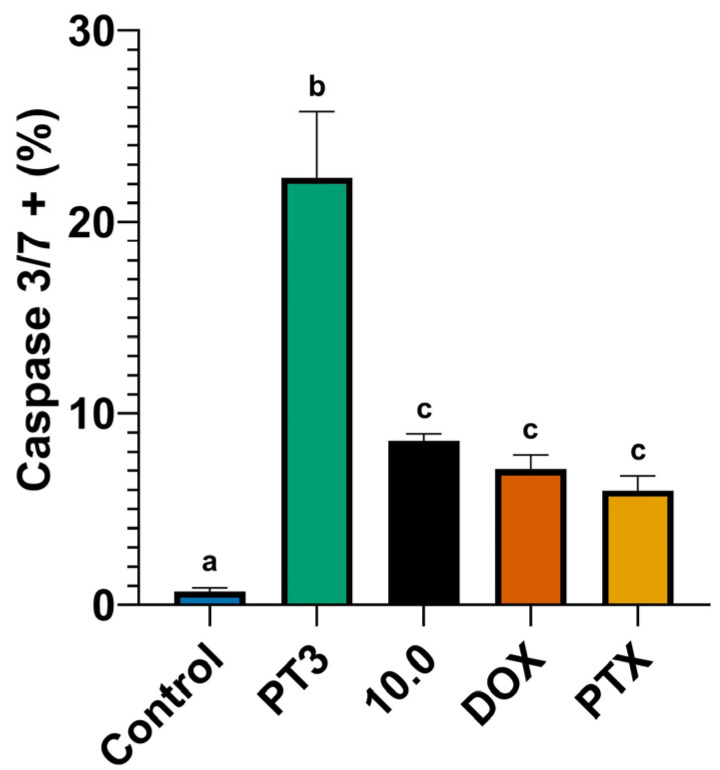
Caspase-3/7 activity in TIB-223 cells treated with tocotrienols (PT3), 10.0 (dietary supplement), doxorubicin (DOX), and paclitaxel (PTX). The graph shows the percentage of caspase-3/7 positive cells in each treatment group. Error bars represent the mean ± SEM from three independent experiments. Different letters (a–c) between treatments indicate significant differences determined by a one-way ANOVA, followed by Tukey’s post hoc test. A significance level (*p*-value) of <0.05 was considered statistically significant.

**Figure 9 nutrients-16-04274-f009:**
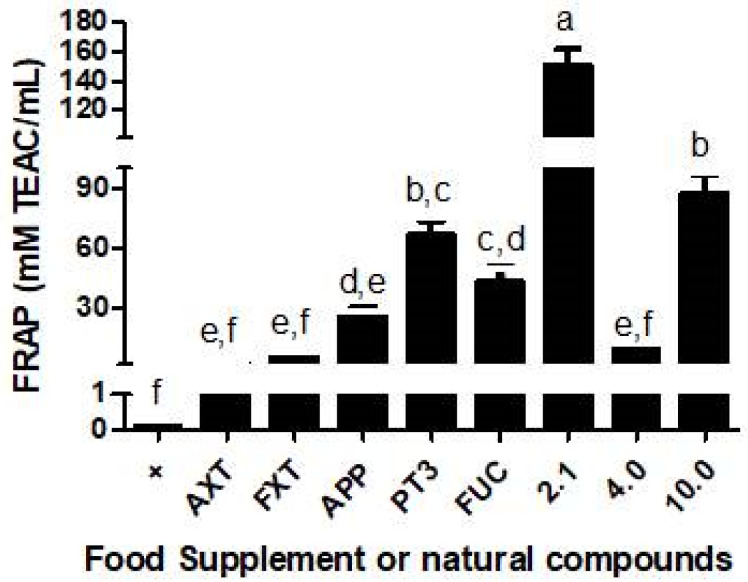
Total antioxidant activity (CAT) of natural compounds and dietary supplements according to a FRAP assay. Different letters (a–f) above the bars indicate statistically significant differences. The data were analyzed by a one-way ANOVA, followed by Tukey’s test. A significance level (*p*-value) of <0.05 was considered statistically significant. Key: + is the positive control (α-tocopherol), AXT (astaxanthin), FXT (fucoxanthin), APP (apple polyphenols), PT3 (plant tocotrienols), FUC (fucoidan). Dietary supplements: 2.1 (algas marinas premium), 4.0 (fucoxanthin special formula), 10.0 (ultra).

**Table 1 nutrients-16-04274-t001:** IC_50_ values (μg/mL) of natural compounds on bone (TIB-223), leukemia (K562), colorectal (Caco2), breast (MCF-7), and prostate (DU 145) cancer cell lines.

	TIB-223	K562	Caco2	MCF-7	DU 145	Detroit 548	Vero
PT3	4.3 ± 1.51 ^a^	29.2 ± 6.38 ^a^	39.89 ± 9.04 ^ab^	46.8 ± 6.26 ^a^	78.76 ± 14.27 ^a^	31.20 ± 7.37 ^a^	162 ± 26.50 ^a^
FXT	14 ± 8.24 ^a^	342 ± 65.20 ^a^	262.6 ± 58.80 ^b^	138 ± 33.48 ^a^	448 ± 32.34 ^a^	195 ± 83.80 ^ab^	466 ± 29.20 ^b^
APP	43 ± 21.40 ^a^	977 ± 20.9 ^b^	259.9 ± 64.20 ^b^	985 ± 64.66 ^b^	910 ± 58.02 ^b^	425 ± 130 ^b^	25 ± 4.15 ^a^
FUC	2524 ± 87 ^b^	4189 ± 103 ^c^	2643 ± 126 ^c^	1885 ± 73 ^c^	1066 ± 70 ^b^	2630 ± 122 ^c^	1130 ± 129 ^c^
AXT	2973 ± 179 ^b^	3862 ± 187 ^c^	2277 ± 184 ^c^	1709 ± 64 ^c^	2265 ± 77 ^c^	2984 ± 102 ^c^	1226 ± 216 ^c^
DOX	1.16 ± 0.21 ^a^	3.25 ± 0.34 ^a^	6.5 ± 0.29 ^a^	0.67 ± 0.10 ^a^	102.60 ± 9.30 ^a^	1.21 ± 0.29 ^a^	1.68 ± 0.60 ^a^
PTX	1.31 ± 0.26 ^a^	125 ± 34.40 ^a^	0.23 ± 0.08 ^a^	11.3 ± 0.39 ^a^	1.49 ± 0.07 ^a^	0.67 ± 0.25 ^a^	5.19 ± 1.17 ^a^

Each column represents the data (IC_50_ values) for a specific cell line treated with each compound (PT3 (plant tocotrienols), FXT (fucoxanthin), APP (apple polyphenols), FUC (fucoidan), AXT (astaxanthin)) for 24 h. Doxorubicin (DOX) and paclitaxel (PTX) were used as positive controls. The IC_50_ values, calculated from the mean ± SEM of triplicate determinations, indicate the concentration required to inhibit 50% of cell viability. The IC_50_ values for each cancer cell line were determined using the MTT method. ^a–c^ Values with different superscript letters within each column are significantly different. All data were analyzed by a one-way ANOVA, followed by Tukey’s post hoc test. A significance level (*p*-value) of <0.05 was considered statistically significant.

**Table 2 nutrients-16-04274-t002:** IC_50_ values (μg/mL) of dietary supplements on bone (TIB-223), colorectal (Caco2), breast (MCF-7), leukemia (K562), and prostate (DU 145) cancer cell lines.

	TIB-223	Caco2	MCF-7	K562	DU 145	Detroit 548	Vero
2.1	1160 ± 121.50 ^b^	340.50 ± 59.13 ^a^	821.2 ± 105.7 ^b^	295.10 ± 59.68 ^b^	1943 ± 175.60 ^b^	191.40 ± 49.82 ^b^	2238 ± 150.30 ^c^
4.0	2302 ± 89.33 ^c^	2373 ± 306.60 ^b^	2296 ± 111 ^c^	7300 ± 18 ^c^	7185 ± 176 ^c^	5391 ± 80 ^c^	8770 ± 247 ^c^
10.0	126 ± 25.32 ^a^	158.70 ± 17.79 ^a^	200.5 ± 44.04 ^a^	243.70 ± 32.90 ^b^	396 ± 10.05 ^a^	166.90 ± 21.38 ^b^	386.30 ± 26.71 ^b^
DOX	1.16 ± 0.21 ^a^	6.50 ± 0.29 ^a^	0.67 ± 0.10 ^a^	3.25 ± 0.34 ^a^	102.60 ± 9.30 ^a^	1.21 ± 0.29 ^a^	1.68 ± 0.60 ^a^
PTX	1.31 ± 0.26 ^a^	0.23 ± 0.08 ^a^	11.3 ± 0.39 ^a^	0.12 ± 0.03 ^a^	1.49 ± 0.07 ^a^	0.67 ± 0.25 ^a^	5.19 ± 1.17 ^a^

Each column represents the data (IC_50_ values) for a specific cell line, treated with different dietary supplements (2.1 (algas marinas premium), 4.0 (fucoxanthin special formula), 10.0 (ultra)) for 24 h. Doxorubicin (DOX) and paclitaxel (PTX) were used as positive controls. The IC_50_ values are reported as mean ± SEM from triplicate determinations, showing the concentration required to inhibit 50% of cell viability. The IC_50_ values for each cancer cell line were determined using the MTT method. ^a–c^ Values with different superscript letters within each column are significantly different. All data were analyzed by a one-way ANOVA, followed by Tukey’s post hoc test. A significance level (*p*-value) of <0.05 was considered statistically significant.

## Data Availability

Dataset available on request from the authors due to privacy restrictions.

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
