# Peer review of "Anticancer Activity of Plant Tocotrienols, Fucoxanthin, Fucoidan, and Polyphenols in Dietary Supplements"

_nutrients, 2024, doi:10.3390/nu16244274_

Round 1

Reviewer 1 Report

Comments and Suggestions for Authors

The manuscript by Lara-Hernandez G. et al, described the potential antitumoral effects of natural compounds derived from plants and algae, as well as dietary supplements formulated with the same compounds, on different human cancer cell lines (bone cancer, leukemia, colorectal cancer, breast cancer, and prostate cancer).

The authors describe deeply the antitumoral effects of the plant-derived tocotrienols (PT3) and supplement 10.0 on TIB-223 (bone cancer cell line) that seems the most sensitive cell line. However the antitumoral effects of these compound should be analysed also on other cell lines such as CaCo2 and Detroit 548 cells.

The statistical analysis is not well represented and explained.

Major revisions:

-          In the paragraph 3.1, table 1 and table 2 the authors described the results obtained from MTT assay without showing graphical representation of these results. The results derived from MTT assay should be included. A dose- and time- dependent response MTT assay should be included

-          A quantitative analysis of morphological changes should be added in figure 1.

-          The authors should test proapoptotic effects and the modulation of cell migration of PT3 and supplement 10.0 on at least two other different cell lines.

-          Authors show that PT3 and supplement 10.0 compounds have a pro-apoptotic effect only in live imaging analysis (figure 4 and figure 6). A quantitative analysis should be performed.

-          Cell cycle analysis should be performed upon treatment with PT3 and supplement 10.0 on TIB-223 and at lest other two cell lines.

-          Trypan blue exclusion assay should be performed upon treatment with PT3 and supplement 10.0 on TIB-223 and at lest other two cell lines to demonstrated the presence of necrosis.

-          Demonstration of mechanisms involved in antitumoral effects of the tested compounds is completely missing.

-          In all the figure the authors should explain the statistic method in a proper way. What do “a,b,c,d…” mean?

    Minor revision:

-  They authors should pay attention to the abbreviation

   Minor English revision

Reviewer 2 Report

Comments and Suggestions for Authors

The publication's objective ID nutrients-3275168 was to evaluate the anti-cancer activity of various dietary supplements purchased from the company BGG (Beijing, China). Some major comments and considerations should be addressed:

The results of an analysis of the profile and content of compounds responsible for the investigated activity should complement this work, which is devoted exclusively to the study of the activity of the purchased dietary supplements. Although the supplements were purchased from BGG (Beijing, China), which initially declares the level of active compound content, it is crucial to verify this claim. I kindly request the use of high-performance methods such as HPLC or UHPLC-MS for this purpose. This is particularly important for maintaining the high quality of articles published in the journal Nutrients, which has a high impact factor of 4.8.

Round 2

Reviewer 1 Report

Comments and Suggestions for Authors

The authors requested an additional 20 workdays to complete the requested revision and present the corresponding results.

This request should be accepted:

1.  To conduct an MTT assay that will evaluate the time-dependent response at 12, 24, 36, and 48 hours on TIB-223 cells, to be included as supplementary results

2. To test proapoptotic effects and the modulation of cell migration of PT3 and supplement 10.0 in Caco-2 and Detroit-548 cells.

3. To  conduct orange-acridine experiments in the Caco-2 and Detroit 548 cell lines

Moreover, some mechanicistic evaluation on the signaling pathways involved in caspase 3/7 activation should be included.

The analysis of cell cycle should be included to elucidate in which phase the cells are blocked and to demonstrate the increase of sub-G1 phase. 

Reviewer 2 Report

Comments and Suggestions for Authors

I would like to express my sincere gratitude to the Authors of the publication for their willingness to expand their research to include HPLC analyses. I kindly request an extension of the indicated 20 days for the preparation of the HPLC results.

Author Response

  We accept the request and thank you for extending the 20 days for the preparation of the HPLC results.

Round 3

Reviewer 1 Report

Comments and Suggestions for Authors

After the revisions the manuscript has been improved. Neverteless, few minor revisions should be addressed.

Fig. S4 Control should be included in the graph

Statistic is still not clear. The authors should specify the significance of the letters added in tables and figures (i.e. p value and vs CTRL)
